# Postoperative Complications Are Associated with Long-Term Changes in the Gut Microbiota Following Colorectal Cancer Surgery

**DOI:** 10.3390/life11030246

**Published:** 2021-03-16

**Authors:** Felix C. F. Schmitt, Martin Schneider, William Mathejczyk, Markus A. Weigand, Jane C. Figueiredo, Christopher I. Li, David Shibata, Erin M. Siegel, Adetunji T. Toriola, Cornelia M. Ulrich, Alexis B. Ulrich, Sébastien Boutin, Biljana Gigic

**Affiliations:** 1Department of Anesthesiology, Heidelberg University Hospital, 69120 Heidelberg, Germany; William.Mathejczyk@gmx.de (W.M.); Markus.Weigand@med.uni-heidelberg.de (M.A.W.); 2Department of General, Visceral and Transplant Surgery, Heidelberg University Hospital, 69120 Heidelberg, Germany; martin.schneider@med.uni-heidelberg.de (M.S.); aulrich@lukasneuss.de (A.B.U.); Biljana.Gigic@med.uni-heidelberg.de (B.G.); 3Department of Medicine, Samuel Oschin Comprehensive Cancer Institute, Cedars-Sinai Medical Center, Los Angeles, CA 90048, USA; jane.figueiredo@cshs.org; 4Public Health Sciences Division, Fred Hutchinson Cancer Research Center, Seattle, WA 98109, USA; cili@fredhutch.org; 5Department of Surgery, University of Tennessee Health Science Center, Memphis, TN 38163, USA; dshibata@uthsc.edu; 6Cancer Epidemiology Program, H. Lee Moffitt Cancer Center and Research Institute, Tampa, FL 33612, USA; Erin.Siegel@moffitt.org; 7Division of Public Health Sciences, Department of Surgery, Washington University School of Medicine and Siteman Cancer Center, St Louis, MI 63110, USA; toriolaa@wudosis.wustl.edu; 8Population Sciences, Huntsman Cancer Institute, Salt Lake City, UT 84112, USA; Neli.ulrich@hci.utah.edu; 9Department of Population Health Sciences, University of Utah, Salt Lake City, UT 84112, USA; 10Department of Surgery I, Lukas Hospital Neuss, 41460 Neuss, Germany; 11Department of Infectious Diseases, Medical Microbiology and Hygiene, Heidelberg University Hospital, 69120 Heidelberg, Germany; Sebastien.Boutin@med.uni-heidelberg.de; 12Translational Lung Research Center Heidelberg (TLRC), German Center for Lung Research (DZL), 69120 Heidelberg, Germany

**Keywords:** colorectal surgery, postoperative complications, gut microbiota, 16S rDNA gene sequencing, inflammation, sepsis, anastomosis insufficiency

## Abstract

Changes in the gut microbiome have already been associated with postoperative complications in major abdominal surgery. However, it is still unclear whether these changes are transient or a long-lasting effect. Therefore, the aim of this prospective clinical pilot study was to examine long-term changes in the gut microbiota and to correlate these changes with the clinical course of the patient. Methods: In total, stool samples of 62 newly diagnosed colorectal cancer patients undergoing primary tumor resection were analyzed by 16S-rDNA next-generation sequencing. Stool samples were collected preoperatively in order to determine the gut microbiome at baseline as well as at 6, 12, and 24 months thereafter to observe longitudinal changes. Postoperatively, the study patients were separated into two groups—patients who suffered from postoperative complications (*n* = 30) and those without complication (*n* = 32). Patients with postoperative complications showed a significantly stronger reduction in the alpha diversity starting 6 months after operation, which does not resolve, even after 24 months. The structure of the microbiome was also significantly altered from baseline at six-month follow-up in patients with complications (*p* = 0.006). This was associated with a long-lasting decrease of a large number of species in the gut microbiota indicating an impact in the commensal microbiota and a long-lasting increase of *Fusobacterium ulcerans*. The microbial composition of the gut microbiome shows significant changes in patients with postoperative complications up to 24 months after surgery.

## 1. Introduction

Postoperative complications are still a relevant problem in the daily clinical practice. Due to the continuous improvement in operation techniques and the high safety standards in anesthesia and intensive care, severe perioperative complications have been reduced over the last decades. The use of new diagnostic approaches such as next-generation sequencing (NGS) gave us new insights in microbial processes that are highly complex [1,2,3,4]. Recent studies have already shown that the composition of the patient’s microbiome has an impact on the individual clinical outcome [5,6]. Especially the gut microbiota is in the focus of postoperative complications following major abdominal surgery due to its influence on host immune responses and the occurrence of infections [7,8]. Our working group already discriminated separate microbial communities of the gut microbiome in patients undergoing pancreatic surgery. Depending on the individual composition, patients showed significant higher levels of c-reactive protein (CRP) and an increased leukocyte count, as well as a more frequent incidence of postoperative complications, associated with a prolonged hospital stay [9]. Due to the widespread use of broad-spectrum antibiotics and the impact of the operative procedure itself, it is not surprising that the microbial composition might be hampered [10,11,12]. However, it is still unclear whether these changes are transient or a long-lasting effect. Thus, this prospective, observational study aimed to examine changes in the gut microbiota as well as its impact on the clinical course after colorectal cancer surgery. With particular interest, we focused on long-term changes in this project because nearly all pre-existing NGS data were measured during the primary hospital stay or within the first six months after surgery. However, patients’ long-term outcome has not yet been evaluated in the focus of postoperative complications.

## 2. Methods

### 2.1. Study Design

The present project used stool samples and data from the German site of the multi-centered international ColoCare Study (ClinicalTrials.gov identifier: NCT02328677), approved by the Ethics Committee of the Medical Faculty at the University of Heidelberg, Germany (Ethical Code: S-134/2016). The following inclusion criteria were applied: patients with a newly diagnosed primary invasive colorectal cancer, from the age of 18 years. Patients were enrolled prior to surgery at the Heidelberg University Hospital, Germany, between April 2016 and June 2018. Stool samples were collected before the operation as well as 6, 12, and 24 months afterwards. A total of 62 patients met the criteria and were included into analyses.

Preoperative stool samples were used to determine the baseline microbiota without operative or pharmacological influence factors, and postoperative samples allowed the evaluation of longitudinal changes in the microbiota. Thirty minutes before the surgical procedure, each patient received a standardized single shot of 1 g sulbactam and 2 g ampicillin. Postoperatively, the study patients were separated into two groups consisting of those who suffered postoperative complications and those who had a noncomplicated clinical course according to the Clavien–Dindo classification. Further, all patients were re-evaluated for survival 24 months after the surgery.

### 2.2. Collection and Storage of Stool Samples

Stool samples were collected at baseline and at 6-, 12-, and 24-month follow-up time points. Participants collected and froze stool specimen at home. Samples were brought in during a visit or sent to the lab with provided freeze packs. The stool samples were stored at −80 °C until further processing. Deoxyribonucleic acid (DNA) was extracted from 200 mg plain stool samples using the PSP^®^ Spin Stool DNA Kit (Stratec, Birkenfeld, Germany) according to the manufacturer’s protocol.

### 2.3. Microbiota Analysis 

16S-rDNA libraries were prepared by amplifying the V4 region as previously published and sequenced in an Illumina Miseq instrument (2 × 300 bp) [5,9]. R package dada2 was used to process raw sequences as following: filtration and trim with the following parameters: maximum ambiguity: 0, number of expected errors for each read: 1, and truncate reads at the first instance of a quality score less than 2. Curated reads were as assemble as contigs and checked for chimera with the default parameters. Ribosomal sequence variants (RSV) were then assigned to taxonomy using the Silva database (version 132). In order to analyze only true biological signal, RSV associated with eukaryotes, archaea, and chloroplasts or detected either in the negative control of the extraction or in the PCR were removed from the analysis. The RSV table was used to calculate descriptive indices for alpha diversity (Shannon index), richness (number of RSV observed), and dominance (relative abundance of the most dominant RSV). A total of 2,627,130 reads in total were used, with a mean number of 15,012 reads per sample. Rarefaction curves for all the samples were evaluated to check if a plateau was reached. Beta diversity measures were performed to examine the differences between the samples based on weighted Unifrac distances. 

### 2.4. Diatery Patterns

Dietary behavior within this study was assessed using food frequency questionnaires and has been described previously by Ulrich et al. and Gigic et al. [13,14]. Patients with high scores for a dietary pattern have a greater tendency to follow the pattern compared with patients with a low score. Labeling of the factor was performed quantitatively using a cut-off of 0.40 of the factor loadings. The dietary pattern data were categorized into approximate appropriate tertiles. 

### 2.5. Statistical Analysis

PERMANOVA was performed to analyze the differences in the structure of the microbiota between the groups. Diversity index association with clinical parameters (time, body mass index (BMI), therapy, and disease severity) was performed with a linear mixed-effect model with patient ID as a random effect to consider inter-individual variation, while inter-groups comparison was performed with a Mann–Whitney *U* test. To compare temporal microbial changes at the RSV levels, we used Deseq2 including patient ID as a co-variable. All statistical analyses were performed with R 3.4.4 and the packages Deseq2, microbiome, and phyloseq.

## 3. Results

### 3.1. Cohort Demographics and Clinical Differences

In total, 62 patients undergoing colorectal cancer surgery were included (Table 1). Thirty-five patients underwent rectum resection and 27 underwent a partial excision of the colon. Postoperatively, patients were classified into two groups consisting of those who suffered from postoperative complications (*n* = 30) and those who had a noncomplicated clinical course (*n* = 32). Fifty-eight patients survived the 24-month observation period. 

### 3.2. Associations between Complication and the Microbiota Structure

Patients with a postoperative complication showed no significant differences in alpha diversity compared to patients without complications at baseline (Figure 1). Overall, we observed a decrease of the Shannon index in both groups after operation; however, the change was only statistically significant in the group of patients with a postoperative complication. After operation, the richness decreased significantly in both groups in a long-term manner, with a significant decrease still visible after 24 months. In the group with complication, dominance was significantly increased at 6 months and 12 months and evenness was decreased after operation at 6 months.

The overall structure of the microbiota was unaffected after the operation in patients without complications, while patients with complications showed a shift in their microbiota at 6 months compared to baseline (PERMANOVA: R² = 0.09, *p*-value = 0.006). After 12 and 24 months, no significant difference to the baseline was observed, indicating a good resilience overall (Appendix A). At baseline, patients in the complicated group had a higher abundance of specific taxa, especially species such as *Akkermansia muciniphila, Butyricimonas sp., and Prevotella sp.,* and from the families Ruminococcaceae and Muribaculaceae (Appendix A).

In patients with a postoperative complication, we observed a significant change in the abundance of a large number of RSVs at 6 months (Figure 2). Most of the RSVs were decreased in their abundance after the operation, indicating a large detrimental impact on the commensal’s population. The majority of those RSVs remained significantly decreased at 12 and 24 months, indicating a long-lasting effect. On the other side, a few species were increased in abundance 6 months after the operation, but all of them resolved to the baseline abundance except *Fusobacterium ulcerans,* which still showed a long-lasting increase after 24 months (Figure 2).

### 3.3. Dietary Patterns and BMI

Three major dietary patterns were identified: a western-type diet characterized by high consumption of red and processed meat, and poultry; a bread-and-butter pattern characterized by high intake of bread and butter, and a fruit-and-vegetable dietary pattern characterized by high intake of vegetables, fruits, and vegetable oils. A variance of 74.6% of the total dietary consumption was explained by these three patterns: The western diet pattern explained 35.2%, bread and butter explained 21.5%, and fruit and vegetable explained 17.9% of the variability (Appendix A).

Since diet is a potential modulator of the gut microbiome, we also correlated the dietary patterns with the changes of the microbiota and the clinical course of the patient. None of the dietary patterns showed a significant impact on each group. Noncomplicated group: Shannon index (western: *p*-value = 0.7384; bread and butter: *p*-value = 0.9087; fruit and vegetable: *p*-value = 0. 897), dominance (western: *p*-value = 0.8018; bread and butter: *p*-value = 0.8798; fruit and vegetable: *p*-value = 0.7724), richness (western: *p*-value = 0.4717; bread and butter: *p*-value = 0.7028; fruit and vegetable: *p*-value = 0.9193), and Pielou index (western: *p*-value = 0.9909; bread and butter: *p*-value = 0. 7519; fruit and vegetable: *p*-value = 0.7937). Complicated group: Shannon index (western: *p*-value = 0.8432; bread and butter: *p*-value = 0.6027; fruit and vegetable: *p*-value = 0.1904), dominance (western: *p*-value = 0.8711; bread and butter: *p*-value = 0.5274; fruit and vegetable: *p*-value = 0.5606), richness (western: *p*-value = 0.837; bread and butter: *p*-value = 0.4988; fruit and vegetable: *p*-value = 0.232), and Pielou index (western: *p*-value = 0.7867; bread and butter: *p*-value = 0.3243; fruit and vegetable: *p*-value = 0.3746).

The BMI in the complicated and in noncomplicated group showed no significant difference. Furthermore, we have not detected any impact on the diversity of the microbiome; only the richness in the complicated group was marginally affected (Shannon index: *p*-value = 0.0813, dominance: *p*-value = 0.2798; richness: *p*-value = 0.0247; Pielou index: *p*-value = 0.2956).

### 3.4. Tumor Stage and Chemotherapy

We did not observe any impact of chemotherapy prior or after surgery on the microbiome structure or diversity. Furthermore, we evaluated the impact of an advanced tumor disease on the microbiome. Therefore, we compared patients with and without metastasis, but microbial structure was not affected.

## 4. Discussion

The composition of the gut microbiome has been associated with the clinical course of patients, following major abdominal surgery. Due to the surgical stress and the use of broad-spectrum antibiotics, a reduction in alpha diversity and richness would not be surprising, especially in the early postoperative period. However, the long-term effects of such changes are still uncertain. Therefore, the present study aimed to examine long-term changes in the gut microbiota and the potential influence on postoperative complications in patients undergoing colorectal surgery.

Prior to surgery, we did not observe any significant differences in alpha diversity, richness, dominance, and evenness, between the patient groups with and without postoperative complications. This indicates an intact microbiota prior to the surgical procedure in both groups. Patients with a noncomplicated clinical course showed no changes in their microbial structure in the postoperative period, whereas patients in the complicated group revealed a shift until the six-month follow-up and a recovery until 12 months after surgery. Both groups showed a significant decrease in richness over the whole 24 months after surgery, with a more pronounced effect in the complicated group. The overall alpha diversity was significantly affected only in the complicated group and did not resolve after 24 months. The decrease in bacterial diversity and richness, together with a flare-up of problematic bacteria like *Pseudomonas aeruginosa* is an already described effect that can also be observed in critically ill patients [5,15]. However, apart from the changes in the microbial community, the long-lasting timespan in which this effect is detectable is quite surprising, because trials with shorter follow-ups indicated a faster recovery of the gut microbiota [16]. The reason for this might be that all patients in our cohort received major abdominal surgery with a partial excision of the bowel. Furthermore, the majority of our patients received an additional chemotherapy. Both could be reasons for a longer lasting imbalance of the gut microbiota.

Previous studies in critically ill intensive care unit (ICU) patients have suggested that also a severe dysbiosis is seen in patients with a complicated course [17,18,19]. We cannot confirm these findings, as well as other studies with perioperative patients [9]. Instead of the described severe dysbiosis, more subtle variations seem to be important in the perioperative cohorts, which is probably due to less severe complications and shorter need for antibiotic treatment.

Anti-infective treatments (e.g., antibiotics) are able to aggravate microbial impairment, paving the way for further surgical site infections like *Carbapenem-resistant Pseudomonas aeruginosa.* Recent data indicate that also the selection of the antibiotic itself has an impact on the individual microbial composition. It was shown that in ICU patients, piperacillin–tazobactam was associated with a lower abundance of potentially protective taxa and an increased risk of *Enterococcus* domination [15]. We were not able to show any effects of a specific antibiotic on the gut microbiota. However, our results may support the assumption that the prolonged use of antibiotics due to a complication might accelerate an overgrowth of problematic species and may remove commensals that compete with these taxa. Patients in the group with postoperative complications showed a large spectrum of species that were significantly reduced compared to baseline. While the consortium of commensals tends to resolve after 24 months, we observed a significant loss in the abundance in several taxa, as shown in Figure 2. Potentially protective taxa like *Faecalibacterium* and *Blautia* were significantly decreased after surgery. *Faecalibacterium species* have been evaluated in probiotics in order to prevent a microbial imbalance in the gut [20,21], whereas *Blautia* has been identified as protective in case of a significant decrease in bacterial diversity together with a flare-up of problematic bacteria such as *Clostridium difficile* [22]. Furthermore, not only a loss of so-called protective taxa, but also a long-lasting increase of *Fusobacterium ulcerans* has been shown. *Fusobacterium ulcerans* is a newer species, isolated from tropical ulcers, and resembles most closely the gastrointestinal species *Fusobacterium varium* [23,24]. It belongs to a group of *fusobacteria* that are generally more resistant to antimicrobial agents compared with the oral strains [25]. This might indicate a selection pressure in the gut microbiota under the prolonged antibiotic therapy in the complicated group.

Interestingly, at baseline, the composition of the gut microbiota showed a higher abundance of specific taxa like *Akkermansia* in the complicated group. Researchers assume that a specific *“pathobiome”*, which is not necessarily associated with a fulminant dysbiosis, might be responsible for increased postoperative complications [26,27]. This is in line with our previous findings. We already showed in patients undergoing pancreatic surgery that specific compositions of the gut microbiota, when they appear before surgery, characterized by a higher amount of *Akkermansia* are more prone to complications [9]. Furthermore, the same effect has recently been detected in patients with gastric cancer who underwent a radical distal gastrectomy [28]. There are two potential explanations for the increased abundance of *Akkermansia*. The mucin layer that protects the epithelial cells in the gut is degraded by *Akkermansia,* and this could make the physiological barrier more vulnerable against pathogen invasion [29,30,31]. This might also be the reason for the significant and impressively long-lasting decrease in alpha diversity within the group with postoperative complications. On the other hand, the pro-inflammatory potential of *Akkermansia* is very weak, and its presence seems to have a protective effect on the epithelium, as shown in vitro [32]. *Akkermansia muciniphilia*, in particular, strengthens the integrity of the enterocyte monolayer and is therefore likely to help protect a previously damaged bowel barrier [32,33]. Thus, a shift in the gut microbiome with an upregulation of some taxa might be another physiological function of the immune system to protect itself against pathogens [34].

There are several limitations that need to be addressed. The study was conducted in a single center and included only patients undergoing colorectal cancer surgery. Moreover, we evaluated only a relatively small patient cohort, and some confounders cannot be fully ruled out, which could potentially influence or complicate the correlation of the clinical complication rates and the changes in the gut microbiome; for example, the heterogeneous types of complications, the different types of operations, and the individual antibiotic treatments (e.g., type of antibiotic and length of therapy) in case of a postoperative complication. Furthermore, the majority of patients in the complicated group suffered only from minor complications that required, in most cases, only an antibiotic treatment. Nevertheless, we have performed several statistic tests to evaluate the impact of potential disruptive factors, for example, the use of chemotherapeutic agents, to ensure a reliable dataset.

## 5. Conclusions

To our knowledge, this the first study in colorectal cancer surgery that shows an association between long-term changes in the gut microbiota and postoperative complications. The microbial composition in patients with complication after colorectal surgery revealed significant structural differences and a long-lasting decrease in a large number of species, and indicates an impact in the commensal microbiota up to two years after surgery.

## Figures and Tables

**Figure 1 life-11-00246-f001:**
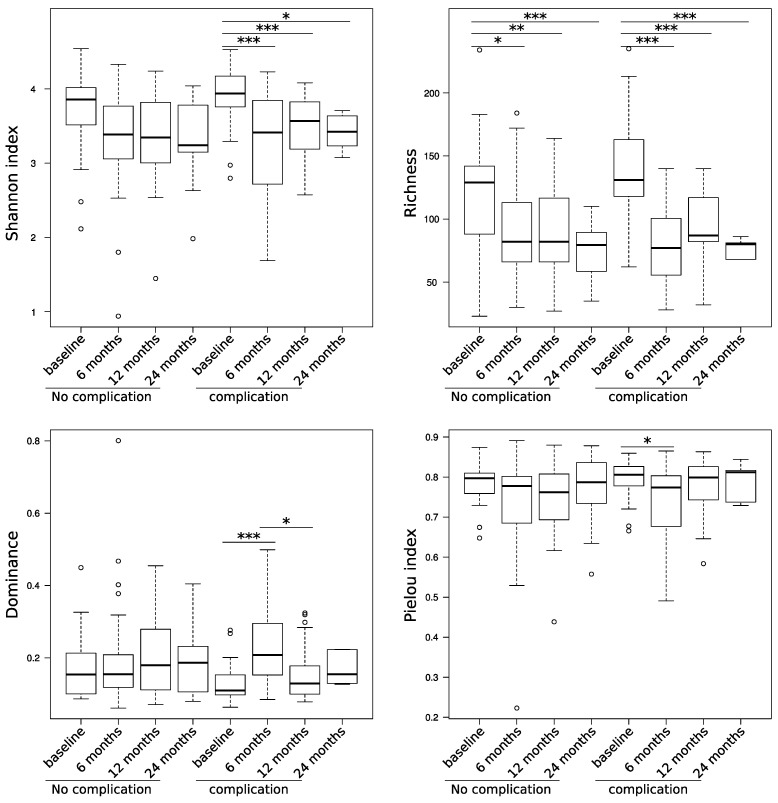
Patients with postoperative complications show a stronger reduction in the alpha diversity, which does not resolve even after 24 months. Following symbols were used to represent significance: *p* < 0.05 was indicated by * *p* < 0.01 by ** and *p* < 0.001 by ***.

**Figure 2 life-11-00246-f002:**
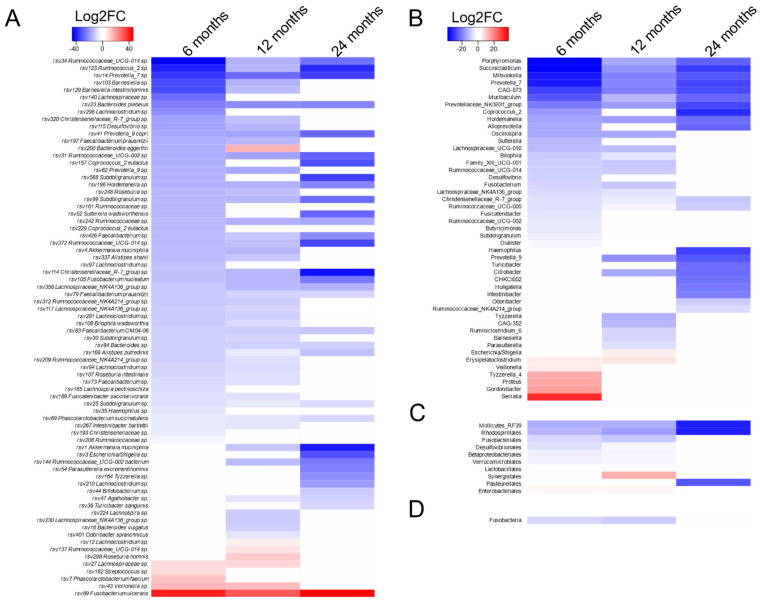
Heatmap representing the longitudinal changes in the group of patients with postoperative complications. Only significantly differentially abundant ribosomal sequence variants (RSVs) (*p*-value < 0.05 and log2 (fold change) > 1) compared to the baseline are displayed (Deseq2). Several taxonomical levels were analyzed: (**A**) RSV, (**B**) genus, (**C**) order, and (**D**) phylum. Taxa enriched in the baseline are displayed in blue, and taxa enriched after complication are in red. The color intensity is relative to the log2 fold change, and nonsignificant taxa are displayed in white.

**Table 1 life-11-00246-t001:** Patients’ characteristics.

Patients’ Characteristics	
**Age (years)**	64 (54.3–72.8)
**Gender**	
Female	34 (54.8%)
Male	28 (45.2%)
**Body Mass Index**	26.4 (24.3–30.0)
**Site of Surgery**	
Colon	27 (58.7%)
Rectum	35 (41.3%)
**Chemotherapy**	
Neo–Adjuvant	15 (24.2%)
Adjuvant	19 (30.6%)
**Tumor Staging**	
Carcinoma In Situ	1 (1.6%)
Grade I	12 (19.4%)
Grade II	18 (29.0%)
Grade III	17 (27.4%)
Grade IV	14 (22.6%)
**Complications (Number of Patients; Double Naming Feasible)**	
Surgical	22 (35.5%)
Medical	20 (32.3%)
Both	14 (22.6%)
**Complication Grade (According to the Clavien–Dindo Classification)**	
Grade I	10 (16.1%)
Grade II	11 (17.7%)
Grade III	7 (11.3%)
Grade IV	2 (3.2%)
**Survivor (After 24-months)**	58 (93.5%)
Data are presented by median and interquartile range (Q1–Q3) or quantity and percentage.	

## Data Availability

The detailed datasets are available in figshare: 10.6084/m9.figshare.14215760.

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
