# Peer review of "Postoperative Complications Are Associated with Long-Term Changes in the Gut Microbiota Following Colorectal Cancer Surgery"

_life, 2021, doi:10.3390/life11030246_

Round 1

Reviewer 1 Report

This manuscript by Schmitt et al. presents a very interesting analysis of gut microbiota in post-surgery CRC patients that experience complications.

Overall the manuscript is very well written, although the results are limited by the small number of subjects involved in the study. Nevertheless, the authors have fully addressed the caveats of the study. I would be curious to see an expanded study in this area based on the current findings.

Author Response

Dear Reviewer No. 1,

We would like to thank the reviewer for the kind feedback!

Best regards,

Felix Schmitt

Reviewer 2 Report

This is an observational study without definite hypothesis where the authors intended to check the long-term effects of abdominal surgery on gut microbiome. Although the theme of the study is interesting, but the following points dampens my enthusiasm:

The discussion is superficially written. From the discussion it is not clear how gut microbiome could be associated with complications? The data is not comprehensively explained. The authors should represent the changes of gut microbiome at phylum, order and genus level. Since diet is the biggest modulator of gut microbiome, how was the diet controlled for the patients during the study? Out of the 62 subjects, what were their characteristics (BMI, clinical chemistry)? The quality of Fig 2 is so poor that I was not able to judge the quality of the result.

Author Response

Dear Reviewer No. 2,

please find attached a detailed point-by-point response.

Best regards,

Felix Schmitt

Reviewer 3 Report

Well written manuscript on interesting topic. I think there are a few issues that could be improved.

  1. While the manuscript is focused on the effect of complications, what a complication is not clearly stated, also with 50% of patients having complications, it must be low threshold, perhaps hiding results . My suggestion would be to define complications as major/minor and focus on CD 3 and 4 complications .
  2. In the discussion it states the … Prior to surgery, we did not observe any significant differences in alpha-diversity, richness, dominance and evenness, respectively between the patient groups with and without postoperative complications...… This is a an important finding, though the study is likely underpowered to come to this conclusion, if   you look at CD 3/4 complications and relooked at this , you may find another result.
  3. Was their any link to the microbiome change and the length of antibiotic use.
  4. As patients with recurrent/metastatic disease often have had more perioperative complications, and have major changes to diet etc, which could influence the microbiome was there any association with recurrent/metastatic disease.

Author Response

Dear Reviewer No. 3,

please find attached a detailed point-by-point response.

Best regards,

Felix Schmitt

Round 2

Reviewer 2 Report

The authors have addressed all of my concerns.